# OpenReview forum: "ABCDE: Agentic-Based Controlled Dynamic Erasure for Intent-Aware Safety Reasoning"
_TMLR — Accepted by TMLR_

### Review · Reviewer_NQvP · 2026-02-09

**Summary Of Contributions:**

This paper argues that most concept erasure methods for text-conditioned generation implicitly assume an unconditional suppression paradigm regardless of context. The authors claim this leads to systematic over-suppression. They propose Intent-Aware Concept Erasure (ICE), which separates (i) whether a concept should be suppressed in a given prompt, and (ii) how suppression is executed. The paper evaluates on a paired benchmark CAEB designed to keep the concept constant while flipping contextual intent, enabling measurement of decision accuracy and reducing confounds from concept identity alone.

**Additional Comments:**

N/A

**Audience:**

No

**Audience Explanation:**

While the proposed benchmark is interesting, the paper’s key observations and technical components appear to be a natural, incremental extension of existing safety-moderation and prompt-rewriting pipelines, rather than a genuine conceptual breakthrough.

**Claims And Evidence:**

Yes

**Claims Explanation:**

The baseline’s high precision is indicative of over-suppression: it achieves few false positives primarily because it aggressively removes the target concept even in benign contexts, thereby avoiding “incorrect preservation” but at the cost of censoring safe prompts. In contrast, the CAEB results demonstrate that the authors’ intent-aware method achieves substantially better overall performance, improving precision while maintaining strong recall, which supports the claim that separating intent decision from execution mitigates unnecessary suppression.

**Requested Changes:**

1.	The paper’s characterization of existing erasure methods as unconditional suppression is not fully accurate. In particular, all model-level safety editing approaches can be considered as intent-aware, in the sense that they change the strength of suppression based on the condition distribution of the input prompt.

2.	While an MLLM may serve as a reasonable oracle in some general settings, relying on it for safety- and ethics-critical, context-dependent judgments is not robust. MLLMs may fail to recognize harmful intent and can reflect or amplify social biases. For example, semantically similar prompts may be treated inconsistently across demographic contexts (e.g., preserving an image of a gun with white people while rejecting it with black people).

3.	The empirical evaluation is limited to the authors’ internally constructed dataset(s), with no results reported on established public benchmarks.

4.	Figure 1 does not clearly convey the core meaning of intent-aware (conditional) concept erasure. The illustrative examples appear to focus on a single benign scenario (e.g., “a kitchen knife in food preparation”), which would naturally fall under “Preserve” and therefore does not adequately demonstrate how intent-aware erasure differs from unconditional suppression in borderline or contrasting contexts.

---

### Review · Reviewer_xx1f · 2026-02-14

**Summary Of Contributions:**

The paper argues that standard concept erasure in text-to-image safety is overly unconditional: it suppresses target concepts whenever they appear, regardless of whether usage is harmful or benign. It proposes Intent-Aware Concept Erasure (ICE), which separates (i) deciding whether to intervene (Remove vs. Preserve) from (ii) executing that intervention. The implementation, ABCDE, is an agentic pipeline with a fixed one-shot intent decision, minimal prompt rewriting, multimodal output verification, and reflection-based execution refinement without revising the original decision. The authors also introduce CAEB, a paired benchmark (Object/Artist subsets) designed to isolate intent by keeping concepts fixed while varying contextual usage.

**Audience:**

Yes

**Audience Explanation:**

I believe  some individuals in TMLR's audience are interested.

**Broader Impact Concerns:**

A broader Impact Statement is included in the paper; it is sufficient to address the ethical implications of the work.

**Claims And Evidence:**

Yes

**Claims Explanation:**

Claims made in the submission are supported by convincing and clear evidence in experiments.

**Requested Changes:**

- CAEB size is moderate (500 + 100 prompts), with explicit focus on clear intent contrasts, which may underrepresent borderline/ambiguous contexts that are central in safety deployment. It might be helpful to further elaborate on why these concepts are important and consider expanding to more concepts.


- ABCDE uses MLLM verification for execution control, while reported CAEB-Object correctness is based on a ResNet-50 threshold proxy. Optimization/selection via one judge and reporting via another can induce hidden mismatch effects, which may introduce biases. It would be more comprehensive to evaluate against several different setups and consider including human verification.


- While prompt templates are provided, key operational specifics (exact hyperparameters for refinement controller, stopping heuristics beyond max K, verifier prompts/config variants per table) are partially described at a high level only. I suggest that authors comprehensively include these details to ensure reproducibility.


- What does ABCDE stand for? I believe this is missing from the current paper.

---

### Review · Reviewer_b6Ne · 2026-02-24

**Summary Of Contributions:**

The paper addresses the issue of over-suppression in generative models, where existing safety mechanisms (concept erasure) remove target concepts regardless of whether the context is harmful or benign.

### Contributions
- A new problem formulation that treats safety as a decision problem grounded in contextual understanding, separating the decision to suppress from the execution of that suppression.
- An agentic workflow consisting of an Intent Analyzer, Minimal Rewriter, MLLM Verifier, and Reflector. It uses iterative prompt rewriting to enforce a fixed safety decision without revising the decision itself during refinement.
- A paired benchmark (500 object-centric and 100 artist-centric prompts) designed to test models on the same concept across different intent contexts (e.g., a chef with a knife vs. a student with a knife) .

### Strengths
- Decoupling the intent reasoning from the generation execution provides a clear, interpretable path for safety interventions.
- Unlike unconditional baselines (ESD, UCE) which systematically fail in preservation scenarios (Precision $\approx$ 0.51), ABCDE maintains high precision ($\approx$ 0.99).

### Weaknesses
- The effectiveness of the refinement loop is heavily dependent on the visual reasoning capabilities of the multimodal verifier, which may have its own biases or blind spots.
- The iterative nature of the agentic loop (LLM reasoning + Image Generation + MLLM Verification) results in significant overhead, with a worst-case runtime of ~40 seconds per sample .

**Audience:**

Yes

**Audience Explanation:**

The paper contributes to several areas of high interest to the TMLR community:
- Safety Alignment: Moving from "dumb" filtering to "smart" contextual reasoning.
- Agentic Workflows: Demonstrating a practical use case for multi-step LLM reasoning in controlling generative outputs.
- Model Editing/Erasure: Providing a framework that operates on the prompt level, avoiding the need for expensive model retraining or fine-tuning.

**Claims And Evidence:**

Yes

**Claims Explanation:**

The authors provide a robust experimental evaluation that supports their claims through quantitative comparison (Table 2 and 5), ablation studies, and cross-model validation.

**Requested Changes:**

- Further clarify the potential for "evasive" prompt rewriting. While the paper mentions "minimal prompt rewriting", it would be beneficial to see more analysis on whether the rewriter might inadvertently preserve harmful intent through stylistic nuances that the MLLM might miss.
- Discuss the scalability of the MLLM verifier. Since the verifier is the "bottleneck" for latency, a discussion on whether smaller, specialized vision-language models could replace large-scale MLLMs would add practical value.
-  Expand the discussion on the $\Delta$ distance constraint. While the authors argue it is a "conceptual constraint", providing a small set of experiments using a concrete metric (e.g., CLIP-text similarity) to quantify the "minimality" of edits would strengthen the utility claims.

---

### Decision · Action_Editor_8wmE · 2026-04-05

**Recommendation:** Accept with minor revision

**Additional Comments:**

The paper addresses the issue of over-suppression in generative models, where existing safety mechanisms (concept erasure) remove target concepts regardless of whether the context is harmful or benign. To be specific, there are several main contributions, including 1)
A new problem formulation that treats safety as a decision problem grounded in contextual understanding, separating the decision to suppress from the execution of that suppression. 2) An agentic workflow consisting of an Intent Analyzer, Minimal Rewriter, MLLM Verifier, and Reflector. It uses iterative prompt rewriting to enforce a fixed safety decision without revising the decision itself during refinement. 3) A paired benchmark (500 object-centric and 100 artist-centric prompts) designed to test models on the same concept across different intent contexts.

However, the authors are encouraged to address the following two concerns in their camera ready version. Please clarify (i) the advantages of the authors’ proposed explicitly intent-aware approach over implicitly intent-aware methods (e.g., fine-tuning that can strength-aware perception intent), and (ii) the reliability of using LLMs as judges. While LLMs can function as a output evaluators, this setting requires them to accurately identify harmful intent, and there has a concern about their robustness. In particular, LLMs may struggle to reliably infer intent, especially given that many jailbreaking attacks succeed by disguising malicious intent.

**Audience:**

Yes

**Audience Explanation:**

The paper addresses the issue of over-suppression in generative models, where existing safety mechanisms (concept erasure) remove target concepts regardless of whether the context is harmful or benign. To be specific, there are several main contributions, including 1)
A new problem formulation that treats safety as a decision problem grounded in contextual understanding, separating the decision to suppress from the execution of that suppression. 2) An agentic workflow consisting of an Intent Analyzer, Minimal Rewriter, MLLM Verifier, and Reflector. It uses iterative prompt rewriting to enforce a fixed safety decision without revising the decision itself during refinement. 3) A paired benchmark (500 object-centric and 100 artist-centric prompts) designed to test models on the same concept across different intent contexts.

**Claims And Evidence:**

Yes

**Claims Explanation:**

The paper addresses the issue of over-suppression in generative models, where existing safety mechanisms (concept erasure) remove target concepts regardless of whether the context is harmful or benign. To be specific, there are several main contributions, including 1)
A new problem formulation that treats safety as a decision problem grounded in contextual understanding, separating the decision to suppress from the execution of that suppression. 2) An agentic workflow consisting of an Intent Analyzer, Minimal Rewriter, MLLM Verifier, and Reflector. It uses iterative prompt rewriting to enforce a fixed safety decision without revising the decision itself during refinement. 3) A paired benchmark (500 object-centric and 100 artist-centric prompts) designed to test models on the same concept across different intent contexts.